# Therapeutic Strategies for Restoring Perturbed Corneal Epithelial Homeostasis in Limbal Stem Cell Deficiency: Current Trends and Future Directions

**DOI:** 10.3390/cells11203247

**Published:** 2022-10-16

**Authors:** Faisal Masood, Jin-Hong Chang, Anosh Akbar, Amy Song, Wen-Yang Hu, Dimitri T. Azar, Mark I. Rosenblatt

**Affiliations:** 1Department of Ophthalmology and Visual Sciences, Illinois Eye and Ear Infirmary, College of Medicine, University of Illinois at Chicago, Chicago, IL 60612, USA; 2Department of Urology, University of Illinois at Chicago, Chicago, IL 60612, USA

**Keywords:** limbal stem cells, limbal stem cell deficiency, limbal stem cell therapy, limbal stem cell transplantation, stem cell therapeutics

## Abstract

Limbal stem cells constitute an important cell population required for regeneration of the corneal epithelium. If insults to limbal stem cells or their niche are sufficiently severe, a disease known as limbal stem cell deficiency occurs. In the absence of functioning limbal stem cells, vision-compromising conjunctivalization of the corneal epithelium occurs, leading to opacification, inflammation, neovascularization, and chronic scarring. Limbal stem cell transplantation is the standard treatment for unilateral cases of limbal stem cell deficiency, but bilateral cases require allogeneic transplantation. Herein we review the current therapeutic utilization of limbal stem cells. We also describe several limbal stem cell markers that impact their phenotype and function and discuss the possibility of modulating limbal stem cells and other sources of stem cells to facilitate the development of novel therapeutic interventions. We finally consider several hurdles for widespread adoption of these proposed methodologies and discuss how they can be overcome to realize vision-restoring interventions.

## 1. Introduction

The cornea is the transparent structure of the anterior eye that plays an imperative role in vision by refracting and transmitting light. This highly specialized tissue is composed of three cellular layers: epithelium, stroma, and endothelium [1]. Composing approximately 10% of the cornea by thickness, the corneal epithelium plays a protective role by shielding more delicate posterior structures and contributing to the stability of the protective tear film [2,3,4]. The corneal epithelium is composed of nonkeratinized stratified squamous cells arranged in approximately 5 layers [5]. Within this barrier separating the anterior eye from the external world, corneal epithelial cells are under constant assault and slough off regularly. Although estimates vary widely, the stratified nonkeratinized squamous epithelium of the cornea may turn over as frequently as every 2 weeks [6]. As with other regenerating epithelial structures, the frequent turnover of the corneal epithelium requires a stem cell population for maintenance. 

The limbus is a narrow region between the cornea and the conjunctiva [7]. Within invaginations of the limbus known as the Palisades of Vogt reside limbal stem cells (LSCs), the cell population required to maintain homeostasis of the corneal epithelium (Figure 1) [8,9]. Although the contribution of LSCs to corneal epithelial regeneration in non-pathological contexts was once debated, we now have compelling evidence that LSCs are entirely responsible for replacing sloughed off corneal epithelium in humans [7,10]. With this recognition of the importance of LSCs, researchers sought to identify specific LSC markers that would facilitate future characterization and isolation of LSCs [11]. To date, numerous LSC markers have been reported, as summarized in Table 1. 

As characteristic of stem cells, LSCs both self-renew and differentiate into rapidly proliferating transient amplifying cells (TACs) that reconstitute the corneal epithelium [29]. LSCs and their immediate descendants from asymmetric division, early TACs, are located in the basal layer of the limbus [22]. Early TACs proliferate and migrate centripetally to populate the corneal epithelium periphery as illustrated in Figure 2. As more mature TACs replace early TACs, these cells lose some of their regenerative capacity [30,31]. Early TACs of the peripheral cornea are capable of a minimum of two rounds of mitosis, whereas more mature TACs of the central cornea become postmitotic after one round of DNA synthesis [30]. Of note, early TACs are also believed to have a reserve of proliferative capacity that is utilized in times of increased demand such as wound healing [22,30]. The ability of LSCs to regenerate the corneal epithelium in both homeostatic and pathological states emphasizes the importance of these cells and implies the existence of specialized regulatory mechanisms that modulate LSC proliferation and phenotype [31]. 

The fate of plastic LSCs is determined through complex interplay between LSCs and their niche. Within their microenvironment, LSCs are exposed to a multitude of inputs including molecular signaling and mechanotransduction events [35]. The cellular populations of the LSC niche are varied and include mesenchymal cells, melanocytes, immune cells, vascular cells, and nerve cells [12]. The presence of specific extracellular matrix constituent proteins also has been widely reported [35], with some research demonstrating that altered ECM interactions may influence LSC phenotype and homeostasis [36]. In their study demonstrating the impact that the microenvironment can have on LSCs, Espana et al. found that rabbit LSCs in the limbal stroma adopt a more quiescent, stem-cell state, whereas LSCs in the corneal stroma are pushed towards differentiation, proliferation, and ultimately apoptosis [37]. Through all the complex interactions of LSCs with their microenvironment, the LSC niche exerts precise control over LSCs, and thus, insults to the limbal niche can have profound impacts on LSC function. After co-culturing LSCs with ultraviolet B (UVB)-irradiated fibroblasts, a resident cell of the limbal niche, Notara et al. [38] found that the presence of inflammatory and proangiogenic cytokines, which were upregulated in response to the UV radiation damage among fibroblasts, ultimately hampered the stemness and colony forming efficiency of LSCs. Therefore, even without direct insult to LSCs, pathological changes to the LSC niche can have deleterious effects on these stem cells [39,40]. For an expert review of pathological alterations to the LSC niche, readers are referred to the review published by Yazdanpanah et al. [12].

If insults to LSCs or their microenvironment are sufficiently severe, a vision-compromising pathological condition known as limbal stem cell deficiency (LSCD) occurs. LSCD is characterized by insufficient corneal regeneration by LSCs either due to the loss or dysfunction of these cells. Subsequent conjunctivalization of the corneal epithelium occurs, leading to opacification, inflammation, neovascularization, and chronic scarring that is a major cause of corneal blindness worldwide [12,41,42]. These pathological changes severely restrict wound repair within the corneal tissue, allowing additional complications such as erosions, ulcers, and potential perforation to develop [35]. Although LSCD has been recognized since the late 1980′s [43], a global working definition for the diagnosis and characterization of this pathology has only recently been established [44]. LSCD can be both a genetic and acquired pathology but it is most commonly attributed to ocular insults from Stevens-Johnson syndrome, burns, and contact lens overuse or misuse [1]. The reported etiologies of LSCD are summarized in Table 2.

## 2. Current Therapeutic Utilization of Limbal Stem Cells

As LSCD progresses and the LSC population is depleted, the most effective therapeutic intervention is transplantation of functioning LSCs. While the initial approach involves harvesting of a sizable sample of limbal tissue from an unaffected eye [53], later strategies have focused on the cultivation and expansion of LSCs for subsequent transplantation [54]. Herein, we first review the existing transplantation methodology and clinical outcomes, and then focus on the current state of ex vivo LSC expansion for transplantation. 

### 2.1. LSC Transplantation

LSC transplantation falls under several categories of transplantation: direct autologous transplantation, in which a limbal graft is harvested from a person’s healthy eye; direct allogeneic transplantation, in which a limbal graft is harvested from a healthy donor eye; cultivated autologous transplantation, in which a small sample of LSCs is harvested, expanded, and transplanted into the same person; and cultivated allogeneic transplantation, in which a small sample of donor LSCs is harvested, expanded, and transplanted [8]. The graft harvested for direct transplantation can be obtained from different tissues, results in the following graft options: conjunctival limbal graft, keratolimbal graft, and simple limbal epithelial [8]. 

Following the discovery that LSCs reconstitute the corneal epithelium, Kenyon et al. demonstrated the first successful cases of limbal graft transplantation for the treatment of LSCD [53]. For the first cases, an autologous graft was used, and in later research has demonstrated varying degrees of success using both allogeneic cadaveric and donor grafts [55,56]. Although allogeneic transplantation can successfully reconstitute LSCs in the limbal niche, the long-term success of an allogeneic graft requires a therapeutic immunosuppressive regime. As the field has progressed, increasingly novel transplantation approaches have been tested [57]. While the autologous approach to transplantation for LSCD treatment has been preferred given that it does not require extensive immunosuppressive medications, this approach is not possible in patients suffering from bilateral LSCD. To confer the benefits of autologous transplantations to patients without an extant population of healthy LSCs, Nakamura et al. demonstrated the feasibility of transplanting cultivated autologous oral epithelial cells on an amniotic membrane substrate [58]. Although they observed mild peripheral neovascularization, all patients reported improved visual acuity at mean follow-up of 13.9 months with noted stability of the corneal surface. 

While the transplantation techniques described above were found to successfully reconstitute the LSC population, a clear hierarchy is apparent among them in terms of the likelihood of clinical success. In their 2020 meta-analysis, Le et al. compiled the results of 40 studies on limbal stem cell transplantation and compared the success rates of direct autologous limbal transplantation, direct allogeneic limbal transplantation, cultivated autologous LSC transplantation, and cultivated allogeneic LSC transplantation [8]. The data reaffirmed that the autologous approach is superior to the allogeneic approach for improving the ocular surface in LSCD. Direct autologous limbal transplantation was associated with the highest rate of ocular surface improvement at 85.7% of cases (95% confidence interval [CI], 79.5–90.3%) followed closely by cultivated autologous LSC transplantation at 84.7% (95% CI, 77.2–90.0%). The success rates with direct allogeneic and cultivated allogeneic approaches for improving the ocular surface trailed at 57.8% (95% CI, 49.0–66.1%) and 63.2% (95% CI, 49.3–75.2%), respectively. Direct autologous limbal transplantation also resulted in the highest rate of improved visual acuity, whereas direct allogeneic limbal transplantation resulted in the highest percentage of cases experiencing recurrent corneal epithelial erosions. These findings collectively suggest that autologous methodologies for LSC transplantation are superior to allogeneic methodologies as they provide better clinical outcomes with fewer adverse effects and do not require treatment with an immunosuppressive regime [59]. Randomized clinical trials comparing these methodologies are still required to elucidate the definitive hierarchy among the limbal transplantation approaches [35]. 

Rama et al. sought to further identify predictors of success in cultivated LSC transplantation and found that the percentage of p63+ cells (presumably LSCs) in culture was correlated with clinical outcomes following transplantation [17]. Consistently, they noted that the number of LSCs in a transplanted culture could predict transplantation outcomes. The transplantation success rate when transplanted cultures consisted of >3% p63+ cells was 78%, that when cultures had >3% p63+ cells was only 11%. Overall, these findings collectively suggest that an autologous approach to limbal transplantation is preferred to an allogeneic approach. Moreover, cultivated LSC transplantation can achieve a similar success rate to direct limbal transplantation but intuitively requires more resources and a larger sample of expanded LSCs to achieve success. 

### 2.2. LSC Culture and Expansion

In cases of uniocular LSCD, autologous transplantation ensures the highest transplantation success rate with a low rate of adverse events. However, harvesting a limbal graft from an unaffected eye is not without risk. Because a sizable limbal graft is required to reconstitute the LSC population, the donor eye is at risk of iatrogenic LSCD [60]. The first reported case of direct autologous limbal transplantation involved the harvesting of a 240° arc of limbal tissue from a healthy eye [53]. In a rabbit model of partial LSCD, when Chen and Tseng surgically removed two-thirds of the limbal epithelium, they observed impaired corneal wound healing suggestive of ineffective repopulation of the corneal epithelium by the remaining LSCs [61,62]. Thus, there seems to be an appropriate intermediary amount of limbal tissue that can be harvested for successful autologous limbal transplantation. Based on the current evidence, limbal tissue grafts derived from less than a 90° arc seem more likely to lead to transplantation failure [63,64], whereas grafts from an arc greater than 240° risk the development of LSCD in the donor eye [65]. These clinical findings are consistent with the findings of Rama et al., intuitively suggesting that an ideal number of transplanted LSCs exists for balancing the likelihood of transplantation success and successful reconstitution of the donor corneal epithelium [17]. 

Because of the risk posed to the donor eye in autologous transplantation, there is increased demand for tissue-sparing approaches. The cultivation and expansion of autologous LSCs provides an avenue to maximize clinical benefit while minimizing the risk of adverse procedural events. Pellegrini et al. [54] reported the first use of cultivated LSCs for autologous transplantation in a LSCD patient in 1997. This approach involves harvesting a small 2 × 2-mm^2^ portion of limbal tissue from the healthy eye for subsequent ex vivo expansion of LSCs [65]. Harvested LSCs are typically cultured for 2–3 weeks on an amniotic membrane scaffold or in suspension after enzymatic processing before transplantation to allow sufficient expansion [66,67]. Although our collective understanding of LSC cultivation has greatly improved over the years, a gold standard methodology that maximizes transplantation success and minimizes the risk of adverse events has yet to be established. As described by Hernáez-Moya et al., many current culture methodologies still rely on animal products and undefined reagents, and these xenogeneic reagents increase the risks of transmitting non-human pathogens and eliciting a host immune response upon transplantation [68]. Therefore, culture practices that do not introduce non-human reagents are urgently needed. Several studies have demonstrated that ex vivo cultivation of LSCs under xenobiotic-free conditions is viable and can translate into favorable clinical outcomes [69,70,71]. As the push to culture LSCs in xeno-free conditions continues, several groups are focused on determining optimal human-based reagent combinations for maintaining LSC stemness and growth ex vivo [68,72,73]. 

In addition to efforts to eliminate the use of non-human reagents in LSC culture, parallel efforts seek to optimize LSC culture conditions and provide an appropriate ex vivo niche for LSC growth. Just as supporting limbal cells are essential for maintaining the LSC microenvironment in vivo, there is clear evidence that supporting feeder cells can enhance the stemness and efficiency of LSCs in culture [74,75,76]. By comparing the absolute number of p63+ cells in culture, Gonzalez et al. determined that the clonal efficiency of LSCs in culture was greater if supporting limbal stromal cells were allowed to maintain cell–cell contact with LSCs than that observed for isolated, single-cell LSC culture [74]. This finding suggests that the supporting cells of the native limbus may retain their supportive niche properties in ex vivo culture, and thus, can serve to maintain LSC phenotype and function. Several other studies have demonstrated consistent findings. While earlier culture systems relied on a monolayer of mouse 3T3 fibroblasts to support LSCs in culture, Nakatsu et al. [76] cultured LSCs in a three-dimensional system on a monolayer of limbal mesenchymal cells and observed maintenance of LSC phenotype in this system. Their approach simultaneously reduced the xenobiotic burden, while their findings demonstrate the importance of supportive cells in culture. Mei et al. and Gonzalez et al. explored similar substitutions and demonstrated that human adipose-derived stem cells and bone marrow stromal cells also may serve as effective feeder cells in a three-dimensional culture platform [75,77]. Other researchers have explored the signaling pathways implicated in the maintenance of LSC stemness in vitro [78,79,80], demonstrating that increasing Wnt signaling, either via inhibition of Wnt inhibitor DKK or introduction of a small-molecule Wnt mimic, promotes LSC stemness and self-renewal capacity [78,79]. Inhibited Notch signaling has also been implicated in maintaining LSC stemness and decreasing proliferation both in the native limbus as well as in vitro culture platforms [80,81,82]. Taken together, these findings emphasize the importance of creating a suitable niche for the growth of LSCs in ex vivo culture. While many of these studies draw inspiration from the native limbal niche for the design of supportive culture systems for LSCs, they may also provide insight into the mechanisms that maintain LSC phenotype and function in vivo. 

## 3. Future Directions: Modulation of Non-Limbal and Limbal Stem Cells to Improve Current Therapies 

Although cultivated and direct autologous LSC transplantation strategies have proven to be effective interventions in unilateral cases of LSCD, bilateral LSCD currently requires an allogeneic limbal graft. Given the benefits of autologous LSC transplantation over allogeneic limbal transplantation, methodologies for autologous transplantation in patients who do not have a healthy reservoir of LSCs are urgently needed. Moreover, allogeneic limbal grafts are often scarce, further limiting this transplantation option [83]. Transdifferentiation of a different stem cell line is one therapeutic avenue that may allow clinicians to avoid allogeneic transplantation in bilateral complete LSCD (Table 3). In the early 2000s, transplantation of cultivated oral mucosa epithelium was found to recapitulate corneal epithelium development in [58], but further research since then has demonstrated that persistent oral epithelium morphology hampers post-transplantation visual outcomes [83,84,85]. Stem cells derived from hair follicle, pluripotent, dental pulp, and umbilical cord sources have been shown to transdifferentiate into corneal epithelium-like cells when exposed to the limbal niche in vitro and in animal models, but related clinical trials have been limited [86,87,88,89,90]. Bone marrow-derived mesenchymal stem cells (MSCs) are a multipotent source that has been validated in human patients with promising results comparable to cultivated limbal epithelial transplantation [91]. As clinical trials of these various stem cells differentiated within the limbal niche are conducted, novel therapeutic avenues that bypass the need for allogeneic transplantation may be realized. 

### 3.1. Molecular Identity and Modulation of LSCs

Another therapeutic possibility for reconstituting the limbal environment is reprogramming existing corneal epithelial cells into LSC-like cells for subsequent cultivation and transplantation. Targeted genetic editing of adult tissues and stem cells has shown promising clinical outcomes in a variety of pathologies and could be applied to LSCD as our understanding of LSCs and the limbal niche advances [92,93,94]. Here we discuss the molecular characteristics of LSCs that are essential to preserving LSC stemness. An understanding of the genes implicated in modulating LSC phenotype and function may simultaneously enable appropriate cultivation methods as well as provide key insights for reprogramming mature corneal epithelial cells into LSC-like cells. 

The Wnt signaling pathway is widely implicated in stem cell self-renewal, quiescence, and differentiation [95]. Profiling of limbal gene expression has revealed preferential expression of Wnt2, Wnt6, Wnt11, and Wnt16b ligands [95,96]. In a co-culture platform with supporting 3T3 feeder cells engineered to express varying amounts of Wnt6 ligand, Bonnet et al. [97] observed that LSCs exposed to high levels of Wnt6 ligand in vitro underwent increased proliferation with decreased expression of terminal differentiation markers of mature corneal epithelium. They suggested that through both canonical Wnt/ß-catenin and noncanonical signaling, Wnt6 may promote LSC self-renewal and stemness. In another study exploring the role of Wnt signaling in LSCs, short hairpin RNA (shRNA) knockdown of Wnt7a in LSCs resulted in expression of skin-specific K1 and K10 with no effects on cellular proliferation [98]. Ouyang et al. elucidated that Wnt7a specifies LSC fate through downstream PAX6 expression. Of therapeutic significance, they also found that skin epithelial cells with upregulated PAX6 expression adopted an LSC-like phenotype and could reconstitute the corneal epithelium in a rabbit cornea injury model. An investigation of chromatin accessibility and the epigenetic landscape of LSCs also demonstrated the role of PAX6 in maintaining corneal epithelium identity [99]. Along with transcription factors RUNX1 and SMAD3, PAX6 forms a core transcription regulatory component, and disruption of this interaction causes LSCs to transition to keratinized epidermal-like cells, consistent with the findings of Ouyang et al. [98]. Taken together, these findings emphasize the importance of Wnt signaling in maintaining LSC phenotype. High Wnt6 expression seemingly promotes LSC self-renewal and stemness, whereas the Wnt7a–PAX6 axis maintains the LSC phenotype. The therapeutic implications of these findings are broad. In a simple application, Wnt6 may be therapeutically employed in LSC cultivation to promote self-renewal and enhance the population of LSCs. Zhang et al. [79] demonstrated such a utility of a Wnt mimic in improving human LSC expansion in an ex vivo system. Furthermore, modulation of Wnt7a expression and downstream PAX6 expression in terminally differentiated skin epithelium also may be utilized to obtain an LSC-like population that can successfully reconstitute the corneal epithelium in LSCD [98]. Although this concept has yet to be applied in human patients, reprogramming of skin epithelial cells into an LSC-like population with subsequent expansion may enable autologous transplantation even in cases of bilateral total LSCD. 

To further understand the roles of Wnt signaling in LSCs, Mei et al. [100] investigated the role of Frizzled receptors in the human limbus. Frizzled receptors are involved in a variety of molecular signaling pathways, including the Wnt signaling pathways. Their qRT-PCR and immunostaining analyses identified the Frizzled 7 (Fz7) receptor as the predominantly expressed limbal Frizzled receptor. Furthermore, immunohistochemistry revealed that Fz7 ligand largely colocalizes with other LSC markers. shRNA knockdown of Fz7 receptor resulted in significantly decreased expression of putative LSC markers as well as decreased colony forming efficiency, a marker of LSC stemness. Together, these results suggest that the Fz7 ligand–receptor interaction promotes LSC stemness and function. Once more, this basic science finding may have profound therapeutic implications. Therapeutic upregulation of Fz7 signaling may promote the expansion of LSCs and reprogrammed LSC-like cells in culture. As the percentage of LSCs is a core determinant of cultivated LSC transplantation success, understanding how modulation of these signaling pathways can promote LSC self-renewal is necessary [21]. Gonzalez et al. [101] explored the role of Jagged 1 (Jag1)-Notch signaling in human limbal tissue and demonstrated that activation of Notch signaling through a recombinant Jag1 ligand decreases LSC stemness and causes differentiation towards mature corneal epithelium. These results suggest that Jag1-Notch signaling is a negative regulator of LSC stemness, and further research exploring the impact of downregulated Jag1 expression on LSC stemness might identify another strategy for promoting LSC expansion. For an expert review on Wnt and Notch signaling activity in the limbus, readers are referred to the review published by Bonnet et al. [95].

Single-cell RNA sequencing (scRNA-seq) technology has enabled the genetic characterization of LSCs and their subpopulations [22,102,103,104,105]. Investigations of differential gene expression along the continuum from quiescent LSCs to mature differentiated corneal epithelium produce key information regarding the roles of various genes in LSC function and differentiation. By identifying subclusters of LSCs ranging from quiescent to actively differentiating, Li et al. [103] identified TSPAN7 and SOX17 as potential markers of LSCs. Knockdown of these proteins results in impaired corneal epithelium homeostasis, whereas their upregulation results in increased expression of putative progenitor cell markers. Accordingly, these two proteins may be markers of LSCs with the capacity to regenerate damaged corneal epithelium. If further validated, these two protein markers may serve as screening markers to support successful cultivated LSC transplantation, potentially improving transplantation outcomes. Through transcription factor expression analysis, SOX9 expression was found to play a role in modulating LSC quiescence and activation [106]. Cytoplasm-to-nucleus shunting of SOX9 seems to drive LSCs into an activated and differentiating state. RNA interference (RNAi) knockdown of *Sox9* in vitro results in increased expression of stem cell and terminal differentiation markers and decreased expression of progenitor cell markers. These results indicate that cytoplasmic expression of SOX9 maintains LSC quiescence, whereas shunting of SOX9 to the nucleus, the transcription factor’s site of action, and results in maturation of LSCs into TACs. Kaplan et al. [22] also utilized scRNA-seq technology in both wild-type and autophagy-deficient mice to explore the roles of PBK, H2AX, and ATF3 proteins in LSC function. Their results suggest that autophagy upregulates the expression of PBK and H2AX, proteins that drive LSCs towards a differentiated corneal epithelium phenotype, and downregulates the expression of ATF3, a transcription factor that seemingly promotes LSC quiescence. Silencing of ATF3 mRNA results in increased cell growth, suggesting that ATF3 regulates LSC proliferation and maintains quiescence. Although the sequencing depth and read length of scRNA-seq experimentation may lead to error and varying results, the application of this technology to the limbal niche has provided insights into genetic markers of LSC quiescence and differentiation [107].

Finally, ABCB5 and Plk3 signaling have also been implicated in regulating LSC proliferation and apoptosis. Ksander et al. demonstrated that transplanted LSCs with ABCB5 positivity can successfully reconstitute the corneal epithelium in animal models of LSCD, whereas ABCB5 knockout results in increased LSC proliferation, apoptosis, and ultimately a LSCD phenotype [21]. They concluded that ABCB5 modulates LSC quiescence and survival through anti-apoptotic signaling, thereby providing another switch that could be modulated in strategies to promote LSC expansion and reprogramming of mature epithelial cells. The hypoxia-induced downregulation of Plk3 in LSCs similarly prevents LSC apoptosis but dissimilarly promotes LSC differentiation [108].

Altogether, this summary of molecular markers of LSC function provides a repository of potential gene targets that may maintain and enhance LSC function (Table 4; Figure 3). The benefits of this are threefold: (1) modulation of these signaling pathways may be applied to increase cultivated LSC stemness and self-renewal, thereby improving the likelihood of clinical success post-transplantation; (2) development of pharmacological agents that harness these pathways to increase LSC self-renewal and proliferation may benefit treatment of partial LSCD, paving the way for improved medical treatment of LSCD; and (3) modulation of either mature corneal epithelial cells into an LSC-like phenotype or the recovery of damaged LSCs in partial LSCD may have profound therapeutic implications. Reprogramming of mature epithelial cells into an LSC population may enable the superior autologous transplantation in cases of bilateral LSCD. While the present review focuses on LSCs and potential strategies for modulating different cell populations, the role of the limbal niche in successful transplantation therapies is also being investigated. The healthy limbal niche can dedifferentiate mature corneal epithelial cells into LSC-like cells [109], whereas a disturbed limbal niche can culminate in LSCD. The therapeutic strategies discussed in this review may be ineffective when applied to a sufficiently damaged limbal niche. As the limbal niche is damaged, an inflammatory process pathologically impacts the function of both resident supporting cells and LSCs [12]. This process chronically remodels and distorts the microenvironmental structure of a healthy limbal niche. Altogether, damage to the limbal niche can be conceptualized as alterations to its functional components including supporting niche cells, the structural extracellular matrix, and soluble biological factors [12]. As these multifactorial disturbances to the limbal niche continue to accumulate, they ultimately overwhelm and impair the functionality of LSCs and culminate in a disease phenotype. Current efforts in bioengineering and reconstructing a healthy limbal niche have been reviewed by Yazdanpanah et al. and Bonnet et al. [12,95]. 

### 3.2. Potentially Underutilized Role of (Lymph)angiogenesis Modulation in LSCD 

Just as modulation of LSC stemness and phenotype in culture and in vivo may lead to improved therapies for LSCD, the ability to regulate pathological neovascularization may provide an additional avenue of treatment. It has been postulated that the limbus acts as a physiologic or physical barrier against invading blood vessels to the cornea after injury [110]. In a previous study where we generated five injury models that involve debridement of the epithelial layer of the limbus, cornea, or both, we assessed the contributions of each epithelia to corneal avascularity. Debridement of the whole cornea resulted in significant blood and lymphatic vessel growth, while that of the whole limbus yielded minimal corneal blood and lymphatic vessel growth. Following hemilimbal plus whole corneal debridement, corneal blood vessel growth occurred only through the non-injured aspect of the limbus. These results indicate that the integrity of the corneal epithelium is important for (lymph)angiogenic privilege. However, the limbus may not act alone as a barrier to invading vessels, but other factors may be involved [111]. Another review suggests that with damage to LSCs and limbal niche, a cascade of inflammatory cytokines induces leukocyte chemotaxis to the site of injury [112]. Release of pro-angiogenic factors, such as vascular endothelial growth factor A (VEGF-A) by macrophages, corneal epithelial cells, and conjunctival epithelial cells results in a disrupted balance between pro- and anti-angiogenic factors that subsequently allows neovascularization [112]. Although corneal neovascularization is a non-specific finding in LSCD, this phenomenon is known to result in edema and persistent inflammation, consequently impairing visual acuity in many corneal pathologies including LSCD [113]. As persistent inflammation continues, the limbal niche increasingly deviates from a healthy phenotype. While topical corticosteroids are commonly employed as a temporizing measure in the early stages of LSCD [114], the anti-inflammatory effects of steroids seem to simply delay neovascularization rather than prevent it [115]. Kadar et al. [115] demonstrated the utility of bevacizumab, an FDA-approved anti-VEGF antibody, for reducing corneal vascularization in rabbit models of LSCD; however, preventive bevacizumab treatment prior to the development of corneal neovascularization does not seem to effectively prevent all neovascularization. Instead, this research revealed an additive benefit of corticosteroids and bevacizumab for reducing corneal neovascularization in LSCD. Further efforts exploring the link between inhibition of corneal neovascularization and LSCD progression are required to confirm the clinical benefit of this strategy. It is possible that this approach may be effective as an additional temporizing measure in early LSCD and may even stabilize the course of the disease.

In addition to their potential role in pre-transplantation LSCD management, therapeutic approaches targeting neovascularization also have utility post-transplantation. While cultured oral mucosa epithelium transplantation was proposed as a promising alternative method to autologous transplantation in bilateral cases of LSCD, this approach is often associated with post-transplantation superficial corneal neovascularization that may ultimately compromise vision [116]. Using immunohistochemistry and confocal microscopy methods, Chen et al. [116] characterized the expression of pro-angiogenic molecules in cornea specimens of patients after cultivated oral mucosa epithelium transplantation. Quantification of corneal neovascularization after transplantation has also been described as a means of assessing post-transplantation success in cases of autologous cultured LSC transplantation, attesting to the importance of this phenomenon in maintaining transplanted limbal graft viability [117]. Zakaria et al. [118] reported a specific function of lymphangiogenesis in cultured LSC transplantation rejection, highlighting the role of lymphatics as the afferent arm of immune rejection and the role of blood vessels as the efferent arm of immune rejection. It is possible that adjuvant treatment targeting neovascularization may improve transplantation outcomes in LSCD. This concept has already been applied with successful outcomes in cornea transplantation, with the cornea graft survival rate improved by administration of anti-VEGF treatment [119].

Our lab has developed dual transgenic murine cornea models that allow simultaneous in vivo observation of angiogenic and lymphangiogenic dynamics in a variety of conditions [111,120]. This platform further enables the development of conditional knockout models in which gene expression can be precisely controlled. We have proposed modulation of angiogenic and lymphangiogenic dynamics as a means to treat pathologies with an inflammatory component [121]. Application of our dual transgenic murine model in conjunction with models of LSCD may be of great use in exploring the benefits of anti-(lymph)angiogenic treatment in both stabilizing LSCD pathology and improving limbal transplantation outcomes. Future endeavors in this regard may yield discoveries that facilitate bench to bedside therapeutic development. 

## 4. Therapeutic Hurdles of LCSD 

While therapeutic applications of cultivated and simple LSC transplants have progressed in recent decades, several hurdles to their widespread availability and adoption persist. Despite numerous clinical trials of LSC-based treatments for LSCD, to our knowledge, only one Phase 4 trial has been conducted. A summary of all upcoming, ongoing, and completed clinical trials is presented in Table 5. 

Until recently, the lack of standardization in both diagnosis and clinical protocols involving LSCs has been a hurdle to the clinical application of the discussed LSCD therapies. Despite its decades-long recognition as a disease entity, global consensus on the definition, diagnosis, and staging of LSCD was only first established in 2019 by the International Limbal Stem Cell Deficiency Working Group [44]; the same group subsequently released a global consensus for the management of LSCD in 2020 [114]. According to this standardization of a LSCD staging system, Le et al. [123] then designed and characterized methodologies to correlate microscopy and imaging findings with in vivo LSC function and LSCD severity. An expert review of current guidelines for LSCD diagnosis and staging is available from Bonnet et al. [35]. 

Despite recent global concurrence regarding the management of LSCD, additional logistical hurdles exist that hamper widespread adoption of both current and future LSC transplantation methodology. Even with evidence that stem-cell based LSCD therapies offer improved treatment options as discussed in Section 3, successful cryopreservation of cultivated LSCs on an amniotic membrane has not been demonstrated [1]. While this may not be a limitation at large medical centers capable of LSC harvesting and expansion, the inability to preserve cultivated LSCs for transport will greatly limit the adoption of this technology in more under-resourced areas. Furthermore, this limitation necessitates a more stringent timeline from LSC harvesting to transplantation, as the LSC amniotic membrane construct must be transplanted while viable. Another logistical obstacle to the application of improved stem cell technologies in LSCD treatment is the financial cost. Angunawela et al. [124] estimated the cost of ex vivo LSC cultivation and subsequent transplantation to be greater than £10,000 in 2010. Thokala et al. [125] described the high costs required to operate and maintain a facility capable of LSC cultivation. With only one laboratory currently offering the expansion of LSCs as a commercial product, cost will likely be a prohibitive factor for the application of improved stem cell-based therapeutics. As discussed in Section 3, an increased understanding of LSC phenotype and molecular function may facilitate the development of novel therapeutics for restoring LSC stemness and the health of the limbal niche. If these therapeutics can be acellular and therefore more amenable to preservation, the development of such a product may address this concern for widespread adoption. Additionally, a therapeutic that restores native LSC function will likely be less expensive than methods for as transplanting cultivated LSCs or LSC-like cells.

Finally, biological hurdles also exist and prevent the widespread adoption of LSC-based therapies and proposed alternatives in LSCD transplantation. One issue that has recently been overcome is the adoption of xenobiotic-free culture conditions in the expansion of LSCs. As discussed earlier in this review, LSCs were originally expanded on 3T3 mouse fibroblasts as supporting feeder cells in medium containing fetal bovine serum [72]. Culture conditions often also involved use of dimethyl sulfoxide (DMSO), a known human toxin and cholera toxin [126]. Gonzalez et al. [72] worked to characterize optimal LSC culture conditions that maintain expansion efficiency while simultaneously adhering to xenobiotic-free practices. The implementation of in-process checks as well as release criteria developed by this group for cultivated LSCs further adhered to Good Manufacturing Practices and regulatory requirements. The standardization and widespread adoption of these safe LSC expansion practices will facilitate the navigation of regulatory processes involved in obtaining FDA approval. Furthermore, the dissemination of these optimal LSC expansion conditions will hopefully facilitate the development of commercial options for LSC cultivation, potentially increasing the accessibility of LSC-based therapies. 

Additional knowledge deficits remain that hinder the development of some of the therapeutic avenues we have discussed, namely (1) improved LSC stemness and expansion ex vivo; (2) pharmacological agents that modulate and restore LSC stemness in a damaged limbus in vivo; and (3) reprogramming of mature corneal epithelial cells into LSC-like cells via cultivation for subsequent transplantation to facilitate autologous transplantation in cases of bilateral LSCD. Further efforts to characterize and modulate the signaling pathways discussed in Table 4 may support the development of pharmaceutical agents that can be easily stored and distributed. Additionally, further characterization of LSCs and their native niche may allow for improved efficiency in cultivation methods as well as efforts to reconstruct a damaged niche. Finally, more thorough characterization of the core genetic and molecular mechanisms that dictate LSC phenotype and function may allow for either (1) genetic reprogramming of adult somatic corneal epithelium into LSC-like cells or (2) restoration of LSC function in a damaged population. For the first approach, issues of somatic retention of age remain when considering the reprogramming of a terminally differentiated tissue [127,128]. Khoo et al. [127] described the dangers of using reprogrammed induced pluripotent cells, which included the retention of genomic alterations, mitochondrial dysfunction, and other characteristics of cellular senescence in the proposed reprogrammed LSC-like cells. The lack of human clinical studies of both this technology as well as transdifferentiation of other stem cells for transplantation into the limbal niche casts uncertainty over the viability of these solutions. 

In summary, several hurdles, both logistical and biological, remain to be addressed for the widespread dissemination of LSC and stem cell technologies for LSCD treatment. A more comprehensive understanding of LSCs and the limbal niche as well as human studies with sufficiently long follow-up periods will help to address and overcome these obstacles.

## 5. Conclusions

LSCs are essential to the maintenance of homeostasis within the constantly regenerating corneal epithelium. LSCD is the vision-compromising pathology that occurs if LSCs or their microenvironment are sufficiently damaged. In this review, we have provided a historical perspective on LSC transplantation and progress that has been achieved with such therapeutic approaches. Despite decades of advancements, improved methods for LSC expansion, modulation, and transplantation are still needed. Based upon the molecular mechanisms within LSCs that have been shown to dictate their phenotype and function, we have identified and proposed several novel therapeutic avenues for modulation of LSCs to achieve (1) improved LSC stemness and expansion ex vivo; (2) pharmacological agents that modulate and restore LSC stemness in a damaged limbus in vivo; and (3) effective reprogramming of mature corneal epithelial cells into LSC-like cells in culture for subsequent autologous transplantation in cases of bilateral LSCD. Recent consensus in both the definition and clinical management of LSCD has certainly paved the way for accelerated therapeutic development, yet certain obstacles remain in the widespread implementation of LSC therapeutics. Standardized methodologies of LSC culture and transplantation that simultaneously maintain xenobiotic-free conditions and prioritize scalability will allow for both safe and efficient dissemination of these therapies. Improved preservation techniques of these cellular therapeutics will also enable long-distance dissemination of LSCD treatment. Finally, an improved molecular understanding of LSCs may also enable the development of acellular therapeutics that can restore function to a patient’s native niche and reverse pathology without requiring surgery. Further characterization of LSCs and translational work with animal models are required to safely adopt these therapeutics in human patients. Through our discussion of several hurdles to global dissemination of these LSC-based treatments, we hope to facilitate the additional work needed for realization of the proposed therapies. 

## Figures and Tables

**Figure 1 cells-11-03247-f001:**
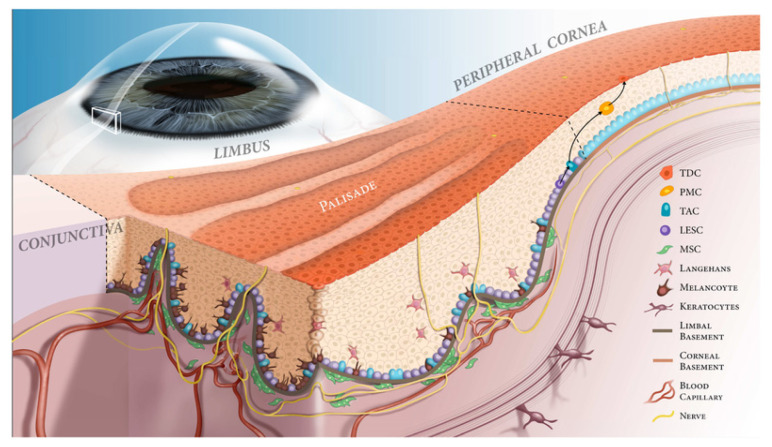
Schematic of the limbal niche. The corneoscleral limbus contains the Palisades of Vogt (PV) and limbal epithelial crypts. The limbal epithelial stem cells (LESCs) are in close contact with niche cells, including melanocytes and mesenchymal stem cells (MSCs) in the limbal epithelial crypts. The basement membranes of the cornea, limbus, and conjunctiva have different constructs, which are in turn necessary for maintaining proper homeostasis. In the basal epithelial layer of the limbal epithelial crypts, the LESCs divide symmetrically into two identical cells (in the horizontal plane) or asymmetrically to give rise to another LESC and a transient amplifying cell (TAC, in both vertical and horizontal planes). Then, TACs divide into postmitotic cells (PMCs) as they migrate centripetally. The PMCs then differentiate into terminally differentiated cells (TDCs) and are shed from the corneal surface.. Abbreviations: LESC, limbal epithelial stem cell, TAC, transient amplifying cell, PMC, post-mitotic cell, TDC, terminally differentiated cell, MSC, mesenchymal stem cell. Reproduced from [12].

**Figure 2 cells-11-03247-f002:**
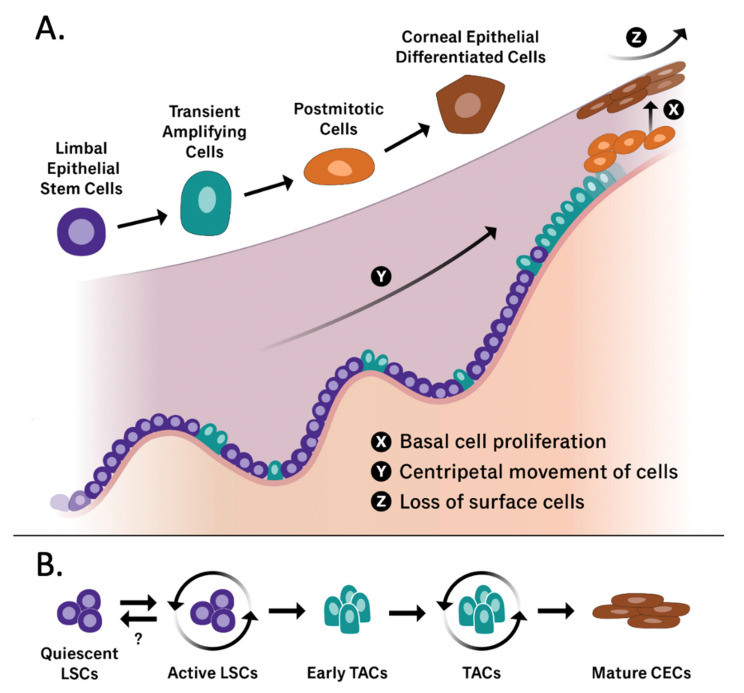
(**A**) Graphical representation of the limbal stem cell niche containing all cell states that have been implicated in corneal epithelium regeneration (LSCs, TACs, postmitotic cells, and corneal epithelial differentiated cells). The X, Y, Z hypothesis published by Thoft and Friend in 1983 [32] is also illustrated, including the three phenomena that allow the corneal epithelial cell mass to remain constant. X: proliferation of basal epithelial cells; Y: contribution to the cell mass by centripetal movement of peripheral cells; Z: epithelial cell loss or constant desquamation from the surface. (**B**) A graphical representation of the differentiation of quiescent limbal stem cells into mature corneal epithelial cells. While the triggers that drive quiescent LSCs into an active differentiating phenotype are well documented, further exploration in reversing this process is warranted. Adapted from [33,34].

**Figure 3 cells-11-03247-f003:**
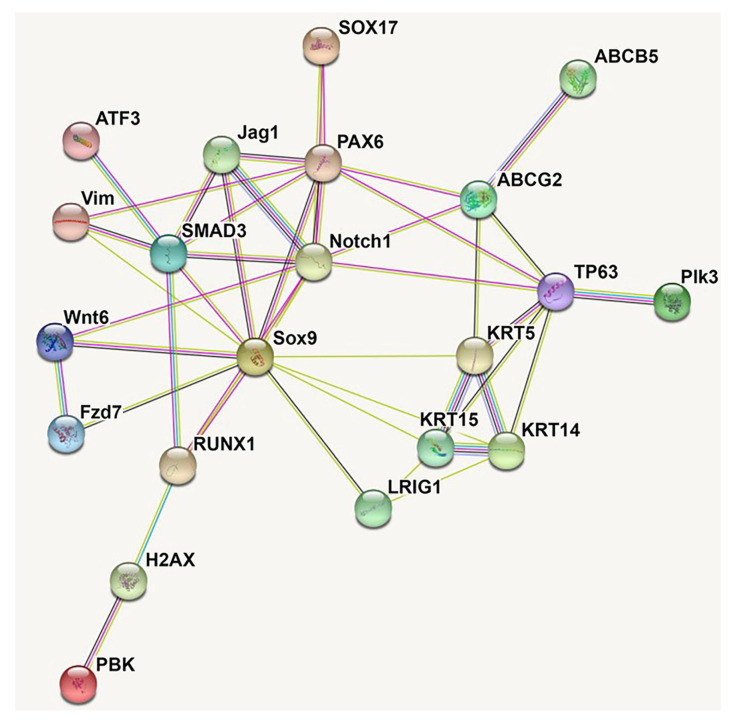
Protein-protein interaction networks of 21 LSC marker genes by STRING analysis. By direct/indirect interactions with other LSC molecular markers, transcription factor SOX9, SMAD3 and PAX6 work together with other stem cell proteins Notch1 and ABCG2, may act as master regulators mediating the proliferation and differentiation of LSC, which is crucial for eye development and corneal surface regeneration.

**Table 1 cells-11-03247-t001:** Putative markers of limbal stem cells.

Marker	Reference	Summary and Application
K15	[13]	Cytokeratin 15 (K15) is preferentially expressed by LSCs in the basal layer of the conjunctival epithelium. In mouse models of LCSD, K15 expression is absent in the limbal epithelium.
K5 and K14	[14,15]	Cytokeratin 5 (K5) and cytokeratin 14 (K14) dimerize and are a main constituent of epithelial cytoskeletons. Chen et al. and Zhao et al. independently observed K5 and K14 colocalization in the basal layer of the limbus. As a marker for LSCs, it is not an entirely specific marker, as differentiated cells derived from LSCs maintain K5/K14 expression.
ΔNp63α	[1,16,17]	The transcription factor p63 was proposed as a marker for LSCs by Pellegrini et al. Later Rama et al. determined that a higher proportion of p63+ cells in an autologous mixed limbal culture transplant correlates with a greater degree of transplantation success. This finding implies that p63 positivity identifies LSCs. However, nuclear p63 expression dictates that other markers must be utilized for enrichment and isolation of LSCs in culture.
ABCG2	[18,19,20]	ATP-binding cassette, sub-family G, member 2 (ABCG2) is a widely expressed stem cell marker. De Paiva et al. identified ABCG2+ cells among limbal basal cells via immunofluorescence staining, and these same cells were shown to have stem cell properties, confirming their identity as LSCs.
ABCB5	[21]	ATP-binding cassette, sub-family B, member 5 (ABCB5) is a known regulator of cellular differentiation. Ksander et al. identified ABCB5 as a marker for LSCs. Transplanted ABCB5+ cells were able to reconstitute the cornea in LSC-deficient mice, whereas *Abcb5* knockout mice had noted depletion of quiescent LSCs.
LRIG1	[22,23]	Leucine-rich repeats and immunoglobulin-like domain protein 1 (LRIG1) expression is proposed to be a marker of epithelial stem cell quiescence. Kaplan et al. utilized single-cell RNA sequencing (scRNA-seq) analysis to identify that a cluster of LSCs and early TACs expressed LRIG1.
TXNIP	[22]	Kaplan et al. observed thioredoxin-interacting protein (TXNIP) expression in the scRNQ-seq cluster corresponding to LSCs and early TACs. They proposed that this protein may contribute to maintaining LSC quiescence through G0/G1 cell cycle arrest.
Notch-1	[7,24]	Notch-1 is a transmembrane receptor that widely regulates cell fate and is implicated in stem cell maintenance. Immunohistochemistry analysis by Thomas et al. demonstrated Notch-1 expression in the limbal basal area, mainly in the PV, as well as overlapping expression of Notch-1 and LSC marker ABCG2, suggesting that Notch-1 may be a promising LSC marker.
C/EBPδ	[25,26]	CCAAT/enhancer-binding protein (C/EBP) δ is a transcription factor implicated in regulating cellular proliferation and differentiation via G0/G1 cell cycle arrest. Barbaro et al. discovered colocalization of C/EBPδ, ΔNp63α, and BMI1 in quiescent LSCs and showed that forced expression of C/EBPδ could exclusively promote self-renewal of LSCs, suggesting C/EBPδ prevents asymmetric division of these stem cells.
Vimentin	[27,28]	Vimentin is the most abundant intermediate filament protein. Vimentin expression has been observed to colocalize with potential LSC markers such as ABCG2 and p63 in the limbal basal area. Although this marker may not provide comprehensive insight on its own, it may prove beneficial in combination with other putative LSC markers.

**Table 2 cells-11-03247-t002:** Major etiologies of LSCD.

Etiology	Pathophysiology and Clinical Context	Reference
Contact lens (CL) wear	LSCD in CL wearers is often asymmetric and bilateral, meaning one eye is more affected than the other. Of the estimated 125 million CL wearers worldwide, roughly 2.4–5% of contact lens wearers develop signs of LSCD. LSC niche damage is hypothesized to be multifactorial, due to hypoxia, mechanical trauma, insufficient lubrication, predisposing factors, eyelid anatomy differences, etc. Presentation of LSCD secondary to CL wear is often initially asymptomatic; subsequent symptoms are generally nonspecific, e.g., pain, photophobia, visual impairment, dryness, irritation.	[45]
Ectrodactyly-ectodermal dysplasia-clefting (EEC) syndrome	An autosomal dominant inherited illness due to heterozygous mutation in the TP63 gene, involving progressive keratinocyte loss, culminating in LSCD and eventual blindness. Ectoderm-derived structures, such as the hair, teeth, skin, and sweat glands, are often compromised. A common defect is anomaly of the Meibomian glands and subsequent instability of the tear film.	[46,47]
Chemical or thermal injury	Most prevalent in males 20–40 years. Non-surgical management (e.g., irrigation, corticosteroids + ascorbic acid, bandage CL, autologous serum, treating high intraocular pressure, tetracycline) to immediately control the inflammation will influence clinical outcomes for the ocular surface and restoration of vision. Sufficient damage to LSCs or niche disarray will culminate in LSCD. After the initial healing process, reconstructive processes can be considered.	[48]
Stevens-Johnson syndrome	SJS is a type IV hypersensitivity adverse drug reaction with a high mortality rate. associated with SJS include anticonvulsants (phenobarbital, lamogtrigine, carbamazepine), antibiotics (sulphonamides, erythromycin, cefotaxime, cloxacillin, quinolones), and non-steroidal anti-inflammatory drugs (NSAIDs). Erythema, erosion, and pseudomembranes affect the oral, ocular, and genital mucous membranes, which may result in insults to the limbal niche.	[49]
Ocular cicatricial pemphigoid (OCP)	An autoimmune ocular disease and type II hypersensitivity response that requires proper management to prevent corneal conjunctivalization, opacification, and irreversible vision loss. Characteristic symptoms included progressive symblepharon (abnormal adhesions between the conjunctiva of the inner surface of the eyelid & conjunctiva of the globe), severely dry eyes, and conjunctival scarring. Twice as prevalent in females vs. males. Topical lubricants for dry eye symptom relief can be used in combination with immunosuppression. The first-line treatment is dapsone, a sulfonamide antibiotic with anti-inflammatory and immuno-modulatory properties.	[50]
LSC transplant donor eye	Great care is taken to preserve the donor eye, which can be a patient’s healthy eye if applicable, or the eye of a suitable donor. LSCD can be induced in the donor eye if excessive limbal tissue is removed. Dissection toward the cornea is extended through the limbal PV to obtain stem cells.	[51]
LSC transplant recipient eye	Measures are taken to limit inflammation and prevent LSC graft rejection. Post-operative management of inflammation involves topical steroids, preservative-free teardrops, and antibiotics. Tarsorrhaphy (stitching the eyelids closed temporarily) after LSC transplantation can decrease susceptibility to dryness.	[51]
Congenital aniridia	Aniridia is a disease in which the iris is partially or completely absent or abnormally developed. Implicated in aniridia-associated LSCD is an abnormality in the PAX6 gene, which has a role in the development of the anterior segment of the eye. Although the corneal epithelium is normal at birth, progressive signs of LSCD become apparent in the range of 20–40 years of age.	[52]

**Table 3 cells-11-03247-t003:** Non-limbal stem cells investigated for the ability to differentiate in the limbal niche.

Cell Source	Summary	Reference
Oral mucosa epithelium	Oral mucosa epithelium transplantation into the limbal area was first reported in 2004. In this work, autologous oral mucosal biopsy samples were obtained, and the submucosal connective tissue was manually removed. The harvested mucosa was divided into smaller sections and the oral mucosa cells were enzymatically separated. Isolated oral mucosa cells were then seeded onto a prepared amniotic membrane with a supporting layer of fibroblasts. After 2–3 weeks, the cultured oral mucosa epithelium on an amniotic membrane were confluent and viable for transplantation. Three eyes afflicted by SJS-induced LSCD and three eyes afflicted by chemical burn-induced LSCD received the prepared oral mucosal epithelium transplants. All cases had improved visual acuity at a mean follow-up time of 13.9 months. Mild peripheral neovascularization was observed in all eyes. Additional clinical trials and studies with similar methodology have been published since this initial report.	[58]
Hair follicle epithelial stem cells	Adult murine hair follicle epithelial stem cells were harvested and cultured in an in vitro environment mimicking the limbal niche. Three- to five-week-old mouse pups were sacrificed and the upper lip pads containing vibrissae were dissected. After enzymatic digestion of the collagen capsule, hair follicles were isolated and underwent further trypsin digestion to isolate individual cells. Hair follicle epithelial stem cells were isolated via FACS. Isolated hair follicle epithelial stem cells were then expanded on a supporting 3T3 fibroblast layer and subsequently introduced to culture conditions mimicking the limbal niche. Limbal-specific extracellular matrix proteins as well as conditioned media from human limbal and corneal fibroblasts were used to emulate the limbal niche. Microscopy, RT-PCR, immunocytochemistry, and western blotting for putative LSC markers confirmed that hair follicle stem cells transdifferentiated into corneal epithelium-like cells under conditions mimicking the limbal niche. A follow-up study by the same group demonstrated an 80% transdifferentiation success rate in an ex vivo mouse model of LSCD. While these methods provide an in vitro concept of transdifferentiation of hair follicle stem cells into corneal epithelium-like cells, this work has not been extended to humans.	[86]
Pluripotent stem cells	A xenogeneic- and supporting feeder cell-free protocol was developed to direct differentiation of human pluripotent stem cells into human LSCs and achieved >65% LSCs in 24 days using either embryonic or induced-pluripotent stem cells. Human pluripotent stem cells were obtained from human embryos and exposed to conditions intended to emulate the limbal niche. In this protocol, Hongisto et al. describe the culture mediums required to induce and differentiate pluripotent stem cells into limbal epithelial stem cells. Furthermore, the authors outline a protocol to cryopreserve and bank the human pluripotent stem cell-derived LSCs, thereby facilitating widespread adoption and dissemination of this technology. This research laid a foundation for subsequent derivation of LSCs from pluripotent stem cells with clear therapeutic implications.	[88]
Dental pulp	Monteiro et al. utilized a chemical-burn rabbit model of LSCD and performed superficial keratectomy 30 days post-injury. Experimental groups then received human immature dental pulp stem cell (hIDPSC) transplants while control groups received amniotic membrane. Immunohistochemistry and RT-PCR analyses showed that hIDPSCs expressed putative LSC markers 3 months after transplantation into the limbal niche. Transplanted hIDPSCs also successfully reconstituted the corneal surface epithelium. To prepare the hIDPSC transplants, the authors first isolated and expanded the stem cells. Three days prior to surgery, the hIDPSCs were lifted and seeded directly onto a temperature-responsive cell culture dish at a density of 2 × 10^6^ per dish. On the day of the surgery, the confluent cell sheets were harvested via a change in temperature and this layer was placed directly on the site of superficial keratectomy and covered with acellular human amniotic membrane. In a similar study, Gomes et al. transplanted a sheet of tissue-engineered hIDPSCs covered by an amniotic membrane into the same rabbit model of LSCD and again demonstrated the ability of dental pulp cells to reconstitute the corneal epithelium when transplanted into the limbal niche. This stem cell source has not been applied in human patients.	[87,89]
Mesenchymal stem cells	In a 2019 clinical trial, Calonge et al. transplanted allogeneic human bone marrow-derived MSCs into the limbal niche and observed MSC transplantation success rate of 76.5–85.7% at 6-month and 12-month follow-ups. They reported no significant difference in the transplantation success rate between allogeneic MSCs and cultivated limbal epithelial cells. This trial demonstrated the viability of MSC transplantation into the limbal niche to treat LSCD.	[91]
Umbilical cord stem cells	In this work, bone marrow was harvested from the iliac crest of allogenic donors. Mesenchymal stem cells were isolated and cultured on human amniotic membrane until 90% confluent. Transplantation involved scraping the corneal-limbal pannus and placing the stem cell amniotic membrane graft cell side down. Transplants were sutured and covered with a bandage contact lens for 4 weeks.	[90]

**Table 4 cells-11-03247-t004:** Genes and proteins implicated in LSC and niche modulation.

Gene	Reference	Function in LSC Niche
RUNX1, PAX6, SMAD3	[99]	Li et al. utilized chromatin accessibility assays and constructed transcription factor interaction networks to elucidate core transcription regulatory circuitries implicated in modulating LSC function. RUNX1 and SMAD3 were found to be important in maintaining the corneal epithelium, and shRNA knockdown of either RUNX1 or SMAD3 notably results in decreased Notch1 and PAX6 expression, which subsequently disrupts LSC phenotype and stemness. In summary, knockdown of either RUNX1 or SMAD3 causes LSCs to transition to keratinized epidermal-like cells. Through modulation of the epigenetic landscape, RUNX1, PAX6, and SMAD3 maintain corneal epithelium identity. RUNX1 is specifically implicated in histone acetylation that increases the transcription of LSC-specific proteins.
TP63, Jag1	[82,101]	Jagged 1 (Jag1) is a protein expressed in human limbal tissue that activates Notch signaling. Gonźalez et al. demonstrated that Notch signaling activation through a recombinant Jag1 ligand decreases the LSC population and drives LSCs towards a mature corneal epithelium phenotype. By arresting mitotic division in culture limbal epithelial cells, Jag1-mediated activation of Notch signaling decreases basal limbal epithelial cell division. Overall, these findings suggest that Jag1-mediated Notch activation decreases LSC stemness, downregulates p63, diminishes the LSC population, and promotes LSC differentiation to a more mature phenotype. Although Ma et al. demonstrated contrary results in 2007, Gonźalez et al. hypothesized this could be due to differing levels of Notch activation due to different delivery systems of Jag1 ligand.
ABCB5	[21]	ATP-binding cassette, sub-family B, member 5 (ABCB5) is a plasma-membrane protein found in humans. Ksander et al. demonstrated that transplanted ABCB5-positive LSCs can reconstitute the corneal epithelium in a mouse model of LSCD. Furthermore, ABCB5 knockout mice demonstrate enhanced LSC proliferation and apoptosis, ultimately resulting in loss of LSCs and perturbed corneal homeostasis characteristic of a LSCD-like phenotype. Overall, these findings suggest that ABCB5 modulates LSC quiescence and survival via anti-apoptotic signaling.
SOX9	[106]	Through transcription factor gene expression profiling, Menzel-Severing et al. suggested that SOX9 is one of the major transcription factors expressed by LSCs. SOX9 is observed in the cytoplasm of basal LSCs and in the nuclei of suprabasal and corneal epithelial cells, suggesting that shunting of SOX9 to the nucleus of LSCs is associated with increased differentiation and activation. Furthermore, increased expression and nuclear localization of SOX9 is found in LSCs undergoing clonal expansion. RNAi knockdown of SOX9 in vitro results in significant upregulation of stem cell and terminal differentiation markers with simultaneously downregulation of markers of progenitor cells. Thus, a delicate signaling balance is believed to exist between SOX9 and Wnt/ß-catenin signaling that determines the fate of LSC quiescence and differentiation. Cytoplasmic SOX9 expression seems to maintain quiescent LSCs, whereas controlled nuclear translocation may promote shunting of LSCs into TACs.
TSPAN7, SOX17	[103]	Li et al. utilized a scRNA-seq platform to identify subpopulations of LSCs that range from quiescent to actively proliferating and differentiating cells. Characterization of the changes in gene expression along this spectrum of quiescence identified TSPAN7 and SOX17 as novel markers of LSCs that may impact stemness and function. RNA silencing of both mRNA products inhibits cellular proliferation and perturbs corneal epithelial regeneration. Activation of these proteins is associated with increased progenitor cell marker expression. Thus, TSPAN7 and SOX17 may be markers of LSCs with the capacity to regenerate and repair corneal epithelium.
PBK, H2AX, ATF3	[22]	Kaplan et al. utilized scRNA-seq in wild-type and autophagy-deficient mice to characterize molecular differences between LSCs, mature TACs, and mature differentiated corneal epithelial cells. Autophagy-deficient mice exhibit altered expression of PBK, H2AX, and ATF3. Overall, autophagy was found to be a positive regulator of LSCs, promoting differentiation and reconstitution of corneal epithelium in wound healing. Autophagy promotes expression of PBK and H2AX, two proteins that seem to promote LSC differentiation and proliferation, and downregulates expression of ATF3, a transcription factor that seems to promote LSC quiescence. Further investigation of the role of ATF3 via siATF3 treatment demonstrated that downregulation of ATF3 results in increased cell growth compared with that of control siRNA-treated cells. An important implication of this work is the potential regulatory role of ATF3 in decreasing LSC proliferation and maintaining quiescence.
Wnt6	[97]	Bonnet et al. utilized 3T3 feeder cells with differential expression of Wnt6 to observe the dose-dependent effect of Wnt6 on LSC proliferation and differentiation. Co-culture of LSCs with supporting cells expressing high levels of Wnt6 results in increased proliferation of LSCs and decreased expression of differentiation markers. In addition to noncanonical Wnt/ß-catenin signaling, Wnt6 was also observed to activate noncanonical signaling in vitro. Bonnet et al. proposed that medium to high levels of Wnt6 expression are essential in promoting LSC self-renewal and stemness, thereby allowing for optimization and modulation of LSCs in culture.
Frizzled 7	[100]	Mei et al. utilized qRT-PCR and immunostaining to profile the expression of various Frizzled receptors in the human limbus, identifying Frizzled 7 (Fz7) receptor as predominantly expressed. Fz7 ligand colocalizes with other LSC markers and not with mature, differentiated corneal epithelium. shRNA knockdown of Fz7 results in significantly decreased expression of LSC markers, such as ABCG2, K14, and ΔNpP63α as well as significantly decreased colony forming efficiency. These results implicate the role of Fz7 in promoting LSC stemness and function.
Plk3	[108]	Wang et al. utilized a hypoxic stress culture platform to study the differential effects of the hypoxia-induced Plk3 signaling pathway on human LSCs and human corneal epithelial cells. Hypoxic conditions seem to promote LSC differentiation via downregulated Plk3 transcription, whereas hypoxic conditions have the opposite effect on mature corneal epithelial cells, resulting in upregulated Plk3 activity with subsequent apoptosis. This research suggests that downregulated Plk3 activity in LSCs as seen under hypoxic stress promotes LSC differentiation and prevents LSC apoptosis.

**Table 5 cells-11-03247-t005:** Ongoing and upcoming clinical trials involving limbal stem cells.

Reference/ID Number	Procedure	Summary	Trial Phase
NCT03957954	Cultivated LSC transplantation	Cultivated LSC transplantation in uniocular cases of severe-to-total LSCD secondary to injury or ocular surgery as compared to scleral CL control	Phase 1 (Ongoing)
NCT03884569	Cultivated limbal epithelial transplantation	Cultivated limbal epithelial transplantation observational study	N/A (not yet recruiting)
NCT03549299	Pharmacological intervention	Study of the efficacy of investigational medicinal product LSC2 topically administered on eye affected by LSCD. LSC2 contains ABCB5-positive LSCs from cadaveric donors.	Phase 1/2a (Active)
NCT02592330	Cultivated autologous limbal epithelial cell transplantation	LSCs from unaffected eye are harvested, expanded and then subsequently transplanted into the affected eye.	Phase1/2 (Active)
NCT01756365	Cultured corneal epithelium graft transplantation	Autologous corneal epithelium will be cultured to produce a graft of reconstructed corneal epithelium for transplantation.	Phase1/2 (Recruiting)
NCT04995926	Labial mucosal epithelium grafting for corneal limbus substitution	Transplantation of autologous labial mucosal epithelium as a substitute for LSCs, as described by Liu et al., 2011 and Choe et al., 2019.	N/A (Enrolling)
NCT03288844	Cultivated autologous LSC transplantation	Multinational follow-up study of the HOLOCORE trial. Determination of long-term safety and efficacy after autologous cultivated LSC transplantation	N/A (Recruiting)
NCT04021134	Allogeneic simple limbal epithelial transplantation	Observational study investigating the effects of allogeneic simple limbal epithelial transplantation	N/A (Recruiting)
NCT04021875	Autologous simple limbal epithelial transplantation	Observational study investigating the effects of autologous simple limbal epithelial transplantation	N/A (Recruiting)
NCT03943797	Cultivated autologous oral mucosa epithelial sheet	Evaluation of the efficacy of a protocol using collagenase instead of trypsin/EDTA to isolate and cultivate oral mucosal epithelial cells	Phase 1 (Recruiting)
NCT03949881	Cultivated autologous oral mucosa epithelial sheet	Evaluation of the efficacy and patient tolerance of cultivated autologous oral mucosa epithelial sheets in bilateral cases of total LSCD with the use of collagenase as opposed to trypsin/EDTA.	Phase 2 (Recruiting)
NCT04932629	LSC application for treatment of superficial corneal pathologies	Transplantation of ex-vivo cultivated allogeneic limbal stromal cells for patients with unilateral superficial corneal scars	Phase 1 (Not yet recruiting)
NCT02886611	Non-therapeutic investigation	Investigation of genotype–phenotype correlation in patients with genetic etiologies of LSCD	N/A (Recruiting)
Completed			
NCT02577861	Holoclar: cultivated autologous LSC transplantation	Validation of the efficacy of Holoclar for patients with moderate-to-severe LSCD secondary to ocular burn at 1 year post-procedure	Phase 4 (Completed)
NCT03226015	Autologous oral mucosa transplantation	Clinical and histochemical results after oral mucosa graft transplantation in eyes with LSCD	N/A (Prospective, Completed)
NCT02649621	Pharmacological investigation	Prospective clinical trial comparing the improvement of LSCD in vivo after use of Amniotic Membrane Extract Eye Drop (AMEED)	Phase 1 (Completed)
NCT00736307	Cultivated autologous LSC transplantation	Evaluation of the efficacy, safety, and long-term outcomes of transplantation of ex vivo cultured LSCs on amniotic membrane for corneal surface reconstruction in cases of severe LSCD	Phase 2 (Completed)
NCT02415218	Cultivated autologous oral mucosa epithelial sheet	Evaluation of the efficacy and safety of cultivated oral mucosal epithelial sheet transplantation for LSCD therapy	Phase 2 (Completed)
NCT03217435	Epithelial allograft transplantation from living-related donor	Comparison of efficacy of femtosecond laser-assisted corneal epithelial allograft from living-related donor vs. limbal conjunctival allograft from living-related donor for ocular surface reconstruction in patients with LSCD	N/A (Completed)
NCT02568527	Simple limbal epithelial transplantation	Pilot safety and efficacy study of a synthetic biodegradable membrane as a substitute for donor human amniotic membrane [122] for LSCD treatment in combination with limbal tissue freshly excised in theatre as a one-stage procedure	N/A (Completed)
NCT03594370	Non-therapeutic investigation	Study of the ability of optical coherence tomography (OCT) analysis to predict the condition of limbal epithelial stem cells as a potential patient-friendly tool to detect limbal conditions	N/A (Completed)
NCT04773431	Cultivated autologous limbal epithelial cell sheet transplantation	Evaluation of the tolerability and safety of LSCD101 (cultured autologous limbal epithelial cell sheet) transplantation in patients with intractable LSCD	Phase 1 (Completed)
NCT01619189	Cultivated LSC transplantation	Prospective, non-comparative monocentric study of transplantation of allogeneic or autologous LSCs cultured on human amniotic membrane [122] with no feeder cells in eyes with total limbal deficiency	Phase 2 (Completed)
NCT04484402	Autologous LSC transplantation	Treatment of inflammatory-dystrophic corneal diseases using autologous LSCs (corneal epithelial stem cells) or adipose-derived MSCs	Phase 2 (Completed)
NCT04552730	Pharmacological investigation	Study of the efficacy and safety of nerve growth factor in the treatment of LSCD associated with neurotrophic cornea	N/A (Completed)
NCT00265590	Non-therapeutic investigation	Study of specific gene changes in patients with aniridia, a disease in which the iris is fully or partially absent, focusing particularly on corneal changes	N/A (Completed)
NCT01237600	Cultivated LSC transplantation	Study to elucidate the appropriate conditions for developing cultivated corneal epithelial grafts and to evaluate transplantation outcomes	Phase 3 (Completed)

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
