# Peer review of "Therapeutic Strategies for Restoring Perturbed Corneal Epithelial Homeostasis in Limbal Stem Cell Deficiency: Current Trends and Future Directions"

_cells, 2022, doi:10.3390/cells11203247_

Round 1

Reviewer 1 Report

The paper "Limbal stem cells and their therapeutic potential for restoring 

perturbed corneal epithelial homeostasis: current trends and future directions" by Masood et al reports on the essential role played by the LSCs for the proper corneal physiology.

I have few comments pertaining to this manuscript.

1)    Legend for Figure 1 page 2 line 57. Please describe the term X in the equation the sum of X and Y is equal to Z. Y are the postmitotic cells PMCs that migrate centripetally, whereas Z are the terminally differentiated cells TDCs that are shed from the corneal surface. I infer from Figure 2 that X is the basal cell proliferation or proliferation of basal epithelial cells as indicated in the legend for Figure 2.

2)    Page 13 lines 382-3. The Authors stated: "The therapeutic strategies discussed in this review may be ineffective when applied to a sufficiently damaged limbal niche." What, in their knowledge and opinion, would be a sufficiently damaged limbal niche? Were they referring to a certain amount in term of overall limbus circumference or surface? Or were they thinking of the deeply affected cellular function of the LSCs?

3)    Page 22 Conclusion. Based on the expertise of the Authors, what would be the best therapy approach to address the most efficiently and safely the LSCD issues?

Author Response

Reviewer 1: “The paper "Limbal stem cells and their therapeutic potential for restoring 

perturbed corneal epithelial homeostasis: current trends and future directions" by Masood et al reports on the essential role played by the LSCs for the proper corneal physiology.

I have few comments pertaining to this manuscript.

1)    Legend for Figure 1 page 2 line 57. Please describe the term X in the equation the sum of X and Y is equal to Z. Y are the postmitotic cells PMCs that migrate centripetally, whereas Z are the terminally differentiated cells TDCs that are shed from the corneal surface. I infer from Figure 2 that X is the basal cell proliferation or proliferation of basal epithelial cells as indicated in the legend for Figure 2.

We thank the reviewer for highlighting this point of clarity. We have revised the legend according to Figure 1 and have kept the description of X, Y, and Z in Figure 2.

2)    Page 13 lines 382-3. The Authors stated: "The therapeutic strategies discussed in this review may be ineffective when applied to a sufficiently damaged limbal niche." What, in their knowledge and opinion, would be a sufficiently damaged limbal niche? Were they referring to a certain amount in term of overall limbus circumference or surface? Or were they thinking of the deeply affected cellular function of the LSCs?

We thank the reviewer for this important comment. We have added an explanation that unpacks our conceptualization of limbal niche damage. A sufficiently damaged niche refers to pathological alterations to the limbal niche cells, extracellular matrix, and biological factors required for proper limbal stem cell function.

3)    Page 22 Conclusion. Based on the expertise of the Authors, what would be the best therapy approach to address the most efficiently and safely the LSCD issues?”

We thank the reviewer for highlighting this oversight. We have discussed some of the remaining obstacles to widespread dissemination of limbal stem cell therapeutics as well as general advice on how to navigate them safely and efficiently in our conclusion.

Reviewer 2 Report

This review provides a detailed overview of the current therapeutic utilization of limbal stem cell deficiency, which is rich and comprehensive. Especially for the treatment of bilateral LSCD is deeply analyzed, and some constructive opinions and thinking are given in this review. There are several concerns however, that need to be addressed. 

(1)    The subtitle of the article is slightly confused and can not clearly reflect the corresponding content of the article. It is suggested to make some modifications.

(2)    It is suggested to describe in detail the following methods for the differentiation of representative cells into limbal stem cells.

Author Response

Reviewer 2: “This review provides a detailed overview of the current therapeutic utilization of limbal stem cell deficiency, which is rich and comprehensive. Especially for the treatment of bilateral LSCD is deeply analyzed, and some constructive opinions and thinking are given in this review. There are several concerns however, that need to be addressed. 

(1)    The subtitle of the article is slightly confused and can not clearly reflect the corresponding content of the article. It is suggested to make some modifications.

We thank the reviewer for highlighting this inconsistency. We have edited our article title to better suit our manuscript. We have changed the header of section 3 to encompass the contents of this section more thoroughly. We have also modified the header of section 4 for clarity. 

(2)    It is suggested to describe in detail the following methods for the differentiation of representative cells into limbal stem cells.”

We thank the reviewer for highlighting this lack of detail. We have rewritten table 3 in section 3 to include pertinent methodology. We believe this important information adds completeness to this section and portrays the different methodologies available to transdifferentiate various cell lines into LSC-like cells.